# Evaluation of the Impacts of Land Use in Water Quality and the Role of Nature-Based Solutions: A Citizen Science-Based Study

**Julia Calderón Cendejas [1,*], Lucía Madrid Ramírez [1,*], Jorge Ramírez Zierold [2], Julio Díaz Valenzuela [2], Martín Merino Ibarra [3] , Santiago Morató Sánchez de Tagle [1] and Alejandro Chino Téllez [1]**

1   Consejo Civil Mexicano para la Silvicultura Sostenible, Mexico City 01070, Mexico; santiago.morato@gmail.com (S.M.S.d.T.); alex.chino1986@gmail.com (A.C.T.)
2   Posgrado en Ciencias del Mar y Limnología, UNAM, Ciudad Universitaria, Mexico City 04510, Mexico; jzierold@provalledebravo.com.mx (J.R.Z.); jdiazv1@gmail.com (J.D.V.)
3   Unidad Académica de Ecología y Biodiversidad Marina, Instituto de Ciencias del Mar y Limnología, Universidad Nacional Autónoma de México, Mexico City 04510, Mexico; mmerino@cmarl.unam.mx
*   Correspondence: julia.calderon@aya.yale.edu (J.C.C.); lmadrid.rmz@gmail.com (L.M.R.)

**Abstract:** The present study explores the impact of different land uses on water quality in a Mexican basin and addresses key mitigation measures, with key measurements made by citizen scientists. The Amanalco-Valle de Bravo Basin reservoir is the major freshwater supply for Mexico City. By measuring physical-chemical and bacteriological parameters in creeks over 21 months and correlating them to land use areas, it was possible to understand the impacts of different land uses (urban, forest, riparian forests, and different agricultural systems) in water quality. The results show that the concentration of *E. coli*, nitrates, nitrites, total phosphorus, total nitrogen, and total suspended solids were higher than the recommended reference levels, and that average oxygen saturation and alkalinity were lower than the recommended reference levels in most sites. The analysis of the Pearson correlation coefficient showed a strong relationship between water pollution and urban and agricultural land uses, specifically a higher impact of potato cultivation, due to its intensive use of agrochemicals and downhill tilling. There was a clear positive relationship between total forest area and riparian vegetation cover with improved water quality, validating their potential as nature-based solutions for the regulation of water quality. The results of the present study indicate the opportunities that better land management practices generate to ensure communities' and water ecosystems' health. This study also highlights the benefits of citizen science as a tool for raising awareness with regard to water quality and nature-based solutions, and as an appropriate tool for participative watershed management.

**Keywords:** water quality; land use; watershed management; citizen science; nature-based solutions; sustainable management

## 1. Introduction

Restoring the ecosystem services provision in basins is key to ensuring a sustainable water supply to growing cities in the world. To do that, it is necessary to deepen our understanding of the dynamics around human activities, land use systems, and their impacts on water quality, as well as the appropriate solutions to these problems.

For this purpose, several studies have investigated the relationship between land use and water quality parameters that have broadened the understanding of the environmental impact of different land uses.

Among the various chemical substances dissolved in water, phosphorus (P) and nitrogen (N) are particularly important for the management of riverine systems. These two macronutrients are essential components of all organisms and are closely linked to the aquatic carbon cycle, determining both the primary production and the microbial mineralization of organic matter in aquatic systems [1].

Agricultural areas have repeatedly been linked to higher levels of nutrients in water. Nazari-Sharabian et al. [2] found that areas with more dominant agricultural land generated more TN and TP. Kandler et al., 2020 [3] have also linked agricultural uses with higher levels of $NO_3$. Nepomuscene Namugize et al. [4] found a relationship between agricultural uses and higher levels of $NH_4$. Gorgoglione et al., 2020 [5] also found a correlation between TP and agricultural uses. Another study in the Dez River basin in Iran found that dry and irrigated farming in that area generated 77.34% and 6.3% of the Total Nitrogen (TN) load, and 83.56% and 4.3% of the Total Phosphorus (TP) load [6].

Urban land uses have also been widely identified as related to higher levels of nutrients. Gorgoglione et al., 2020 [5] found a correlation between nitrogen concentration and urban uses. Since sediment transport usually plays a significant role in the mobilization of nutrients from urban impervious surfaces, Gorgoglione et al., 2019 [7] confirm that TSS can be considered as a synthetic index of the general level of pollution in urban areas.

Besides identifying the impact of land uses in water, studies have also been able to confirm the provision of ecosystem services from forests, such as water quality regulation, by linking them with lower levels of nutrients in water. Kandler et al. [3] found significantly lower levels of $NO_3$ in forested areas, and Gorgoglione et al. [5] and Nepomuscene Namugize et al. [4] found an opposite correlation between forests in the catchment and TP in water.

Based on the previous studies, this work aims to strengthen the knowledge about the relationship between land uses and water quality by adding another variable that has not been considered in these previous works. This study adds the variable of the production system; hence, it does not only analyze agricultural land as a whole, but the different agricultural systems in the landscape, including traditional rainfed corn cultivation, and more industrial crops, such as fava bean cultivation in irrigated lands and potato crops. This additional variable was analyzed because of the scale of the research, the abundance of water samples collected thanks to the involvement of community members and volunteers, and the possibility to identify the total areas with each crop in every monitored catchment area.

Consequently, the objective of this study is to provide insights to answer the following questions:

(i)    what is the relationship between water quality, land use, and production systems in the Valle de Bravo basin?
(ii)   which water quality parameters are more affected by particular land use categories and production systems?
(iii)  what are the effects of certain nature-based solutions, such as increasing forest cover or restoring riparian vegetation, on water quality?

The research was part of a project led by a Mexican Non-Governmental Organization (NGO)—the Mexican Civil Council for Sustainable Silviculture (CCMSS)—that aimed to improve local capacities for water monitoring and increase awareness about global threats and sustainable solutions. The project allowed for thorough field data collection through citizen science-based water quality monitoring, and a robust analysis in a scientific lab in a national university. Moreover, the project raised awareness and connected water users in central Mexico with the ecosystems that provide them with freshwater, as well as with the communities that protect and manage these ecosystems.

The present study is expected to contribute valuable knowledge for defining effective management strategies to minimize stream pollution through a citizen-based monitoring strategy, driving a community highly involved in both data collection and decision-making.

## 2. Materials and Methods

### 2.1. Study Area

The Valle de Bravo (VB) reservoir receives water from a catchment area of 531 km$^2$, which is the Valle de Bravo basin [8]. It is one of seven reservoirs that are a part of the Cutzamala system, which is a complex of infrastructure that is used to store, pump, purify,

and distribute water, and is one of the main sources of drinking water [8] to Mexico City, Toluca City, and their metropolitan areas, providing water to 13 million people in central Mexico [8]. The Cutzamala system reservoirs are located in two states—Michoacán and Mexico state—and their water is pumped from those reservoirs up to the "Los Berros" potabilization plant before being sent for distribution to the cities.

The VB basin captures around 974 million m$^3$ of water per year, from which 48% returns to the atmosphere through evapotranspiration, 35% is infiltrated, and 17% runs through rivers as surface water [9] that fills the VB reservoir to provide water to cities. Furthermore, at least 841 water springs [9] as well as the basins' rivers provide water to the local population for domestic use, trout production, and irrigation of crops, including maize, fava beans, oats, vegetables, and fruits.

Water quality in the VB reservoir has been declining progressively over time. Human activities in the watershed, including sewage disposal and unsustainable agricultural practices, have affected the water quality of the reservoir since the late 1980s [10]. Nutrient loading to this reservoir increased 276% for phosphorus (P) and 203% for nitrogen (N) in a single decade [11], and a comparative examination of P and N mass balances showed that most (85%) of the P input to VB accumulates in sediments [12]. Recent assessments confirmed eutrophic conditions and cyanobacteria blooms in VB [13,14], with events of high cyanotoxin production (>1.5 μg/L) during the stratification period [15]. The consequences of the level of pollution of VB is seen through impacts in the local populations' health and in the quality of irrigated agricultural products. It also increases the cost of water filtration to produce drinking water, reduces cultural services enjoyed by inhabitants and visitors in the lower basin, and affects economic activities related to tourism [9].

The traditional approach to solve water quality issues has been by filtering and purifying the water from the Cutzamala System reservoirs before sending it for use in central Mexico; however, potabilization costs have become extremely high [16]. Moreover, this approach does not solve pollution problems in the rivers and in the reservoirs, or its consequences for the ecosystem, local population's health, tourism, and the economy. Several studies [17–20] show that restoring the ecosystems by providing water regulation services not only is more cost effective, but it also provides additional benefits, such as biodiversity conservation, carbon sequestration, pollination, generation of livelihoods, and the increase in quality of life for people in the upper basin.

This basin was selected as a case study because of its strategic importance to provide drinking water to the most populated area in Mexico. The study intends to generate recommendations to improve water management policies in this area, as well as in other strategic basins that provide drinking water to large populations.

### 2.2. Citizen Science-Based Water Monitoring Methodology

A total of 165 volunteers from HSBC offices from Mexico City, Toluca, and Guadalajara participated in two-day events during 2018 and 2019. Participants were trained to collect water quality data using the Global Water Watch kit [21] to monitor physical and chemical water quality parameters, as well as a protocol designed by the ABL-UNAM to collect samples for nutrient analysis. The training also included familiarizing volunteers with concepts such as ecosystem services, landscape management practices and their impact on water quality, environmental threats of climate change and urbanization, sustainable development goals, the circular economy, and corporate sustainability.

Additionally, six local team leaders were trained in water monitoring methodologies. This allowed team leaders to also train and guide volunteers during the events, and to monitor water quality during gap months that lacked formal monitoring events.

Volunteers and local team leaders monitored 18 sites in the middle-upper basin over 18 months, assessing 34 water quality parameters.

The "Alabama Water Watch" LaMotte Kit [21] was used to measure the following physical-chemical parameters: water temperature, pH, alkalinity, hardness, dissolved oxygen, and turbidity.

Stream flow was obtained by measuring the area of water in the channel cross section, and measuring the average velocity of water in that cross section using a float in parallel to the water quality monitoring activity.

Samples for bacteriological parameters (*E. coli* and other coliforms) were collected by the citizen scientists. Collected samples were safely transported to the CCMSS's office in Amanalco, where they were incubated in Coliscan EasyGel, which detects a coliform concentration distinguishing between *E. coli* and other coliforms for 30 h to 48 h at 29 °C to 37 °C to be analyzed.

Additionally, water samples were sent to the ABL-UNAM to be analyzed for nutrient content ($N-NH_4^+$, $N-NO_2^-$, $N-NO_3^-$, and soluble reactive phosphorus (SRP)). The samples were filtered with 0.22 μm (MilliporeTM type HA) nitrocellulose membrane filters and fixed with chloroform. Analyses were conducted with a Skalar San Plus segmented-flow analyzer using standard methods [22] and specialized analytical circuits [23]. Samples for total nitrogen (TN) and phosphorus (TP) were analyzed for $N-NO_3^-$ and SRP after high-temperature persulfate oxidation [24].

### 2.3. Water Quality Reference Levels

For each parameter, we identified scientific literature or official regulatory instruments (e.g., Mexican Official Norms for water quality) as references for acceptable water quality levels for human contact and the health of aquatic ecosystems.

### 2.4. Monitoring Sites Selection

This study focused on two main sources of pollution: agriculture and urban settlements. Agriculture was divided into (i) maize, (ii) oat, (iii) fava bean, and (iv) potato, which are the main crops of this region. Maize is the most harvested crop in the basin and is used by farmers for self-consumption. Oat is harvested mainly for foraging purposes. Fava bean is an irrigated crop harvested mostly for sale. Potato crops have been promoted by big companies' intermediate buyers in the basin in recent years, who rent land from local farmers and develop the whole production process with a high use of agrochemicals. Besides, land renters usually use straight-line planting in rows that are parallel to the slope to increase runoff and reduce humidity to prevent fungus infections. This practice causes soil erosion and movement of sediment towards the water courses (see description of agricultural cycle and products used in Annex A and B). Regarding human settlements, wastewater has been identified as a major contamination source in the basin [9].

Furthermore, the study assessed the correlation between total forest area and forest cover percentage in the riparian buffers of the catchment areas with water quality parameters to understand the role of nature-based solutions to improve and maintain healthy water courses.

Moreover, 18 monitoring sites were set in the sub-basin of the Amanalco river, which is the main tributary to the Valle de Bravo reservoir (Figure 1). Each site had influence of several different land uses, but some of them had more representation in some of the crops (Table 1).

Additionally, one site in the Amanalco river located upstream of the water discharge of the wastewater treatment plant (PLTR1), and one site downstream of the discharge (PLTR2) were selected to test whether treated wastewater was affecting water quality in the river, as well as to have more information on the treatment effectiveness of the plant.

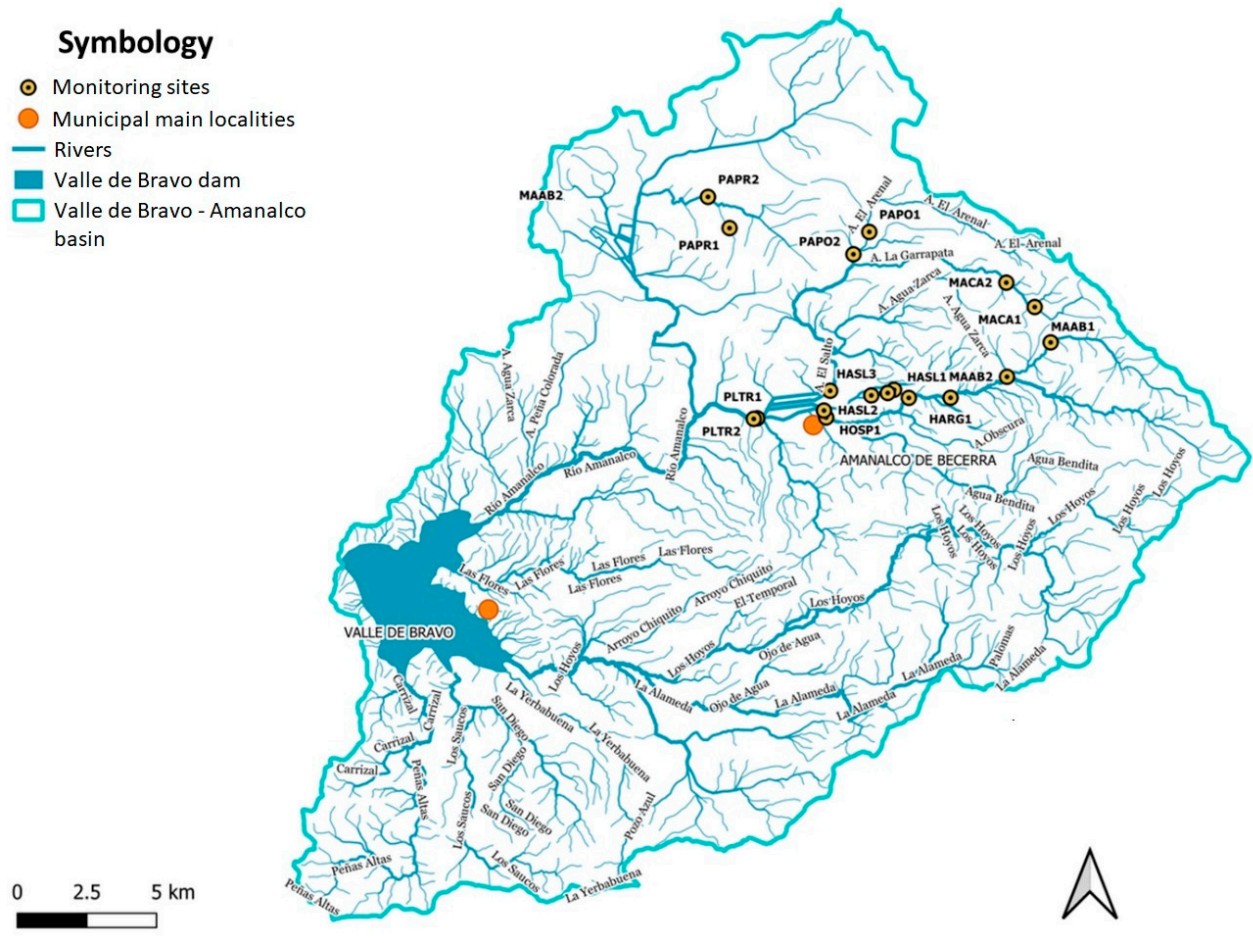

**Figure 1.** Monitoring sites map. The outer blue line shows the limits of the Valle de Bravo-Amanalco basin. The blue area shows the Valle de Bravo dam, and the blue lines show rivers in the basin (width of lines indicates order). The orange dots show municipalities' main localities. Monitoring sites are shown in yellow-black dots.

**Table 1.** Monitoring sites' basic data and number of sample collection times for each group of parameters (*n*B, number of sample collection times for bacteriological parameters; *n*PC, number of sample collection times for physical-chemical parameters; and *n*N-SS, number of sample collection times for nutrients and suspended solids).

| Target Land Use of Influence to the Study Sites | Study Site | Latitude | Longitude | *n*B | *n*PC | *n*SS |
|---|---|---|---|---|---|---|
| Maize | MAAB1 | 19°16′46.26″ | 99°55′23.64″ | 0 | 18 | 2 |
| | MAAB2 | 19°16′42.7″ | 99°56′19.46″ | 21 | 21 | 18 |
| | MACA1 | 19°17′23.77″ | 99°56′39.05″ | 0 | 19 | 18 |
| | MACA2 | 19°17′51.99″ | 99°57′13.95″ | 21 | 21 | 18 |
| | HARG1 | 19°15′38.61″ | 99°58′22.40″ | 0 | 19 | 18 |
| | HARG2 | 19°15′38.47″ | 00°59′13.68″ | 21 | 21 | 18 |
| Fava bean | HASL1 | 19°15′47.83″ | 99°59′31.83″ | 0 | 20 | 18 |
| | HASL2 | 19°15′44.08″ | 99°59′39.58″ | 15 | 21 | 18 |
| | HASL3 | | | 7 | 8 | 8 |
| Potato | PAPO1 | 19°18′50.79″ | 100°00′2.20″ | 0 | 19 | 18 |
| | PAPO2 | 19°18′24.90″ | 100°00′21.71″ | 21 | 21 | 18 |
| | PAPR1 | 19°18′55.22″ | 100°02′53.84″ | 1 | 19 | 17 |
| | PAPR2 | 19°19′31.40″ | 100°03′20.34″ | 21 | 21 | 1 |
| Wastewater from the hospital and human settlements | HOSP1 | 19°15′24.9″ | 100°00′40.39″ | 12 | 12 | 11 |
| | HOSP2 | 19°15′23.78″ | 100°00′57.92″ | 22 | 22 | 18 |
| | SALT | 19°15′46.66″ | 100°00′50.29″ | 14 | 14 | 7 |
| Treatment Plant discharge | PLTR1 | 19°15′14.73″ | 100°02′19.68″ | 32 | 32 | 17 |
| | PLTR2 | 19°15′14.09″ | 100°02′24.18″ | 32 | 32 | 18 |

### 2.5. Characterisation of Land Use in the Catchment Area of Each Monitoring Site

The monitoring sites were selected to represent the influence of predominant land use (Table 1). Nevertheless, none of them were purely influenced by one specific land use, but by a mix of them. Therefore, to assess the impact of each specific land use we quantified the proportion of each land use in the catchment areas or micro-basins that influenced the monitoring sites, so that we could correlate land use area values with water quality parameters.

The quantification process first included delimiting the micro-basins that were influencing each monitoring site through runoff. The delimitation of the influence areas was done using the plug-in GRASS for QGIS 3.10 [25,26]. Figure 2 shows an example of the selection process to define the micro-basin area influencing each monitoring site. Since GRASS calculates drainage using digital elevations models, the resulting run-to-point shapefiles excluded drainage modifications caused by humanmade structures, such as roads or water ditches. To minimize error, a ground recognition of major ditches diverting drainage was performed, and the shapefiles given by the plugin were corrected by deleting areas where water was diverted from those ditches.

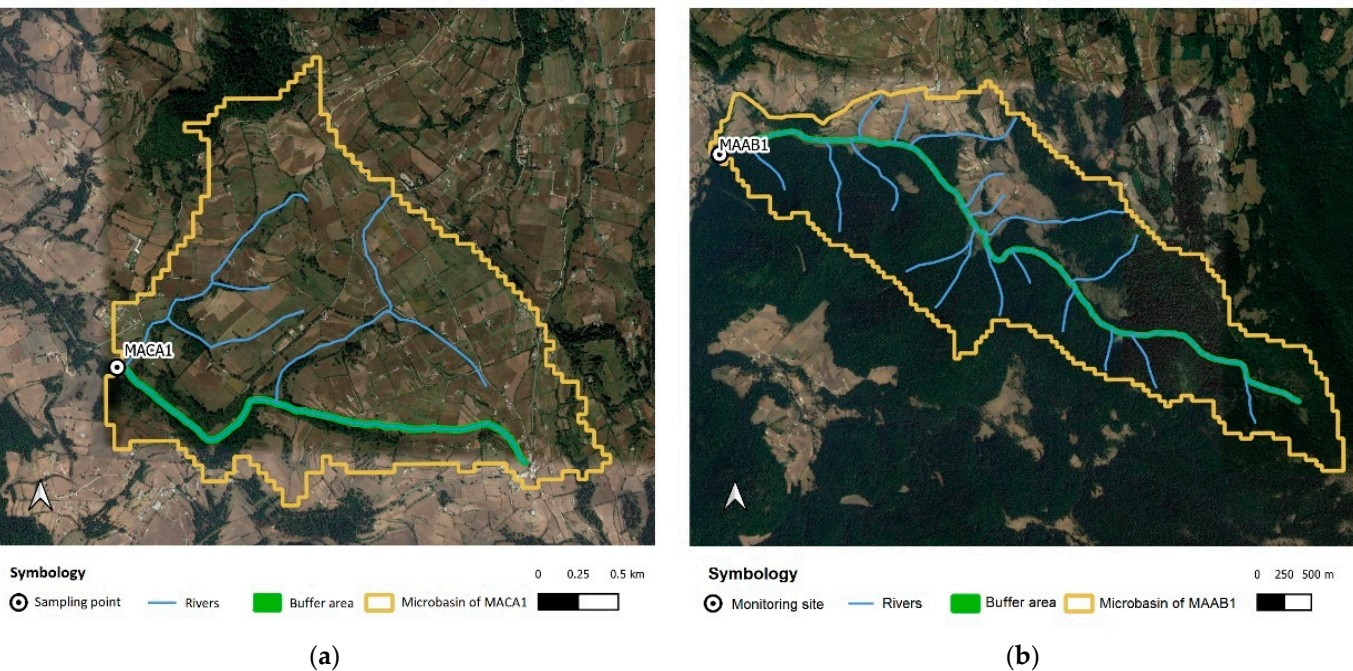

(**a**)　　　　　　　　　　　　　　　　　　(**b**)

**Figure 2.** Examples for the micro-basin's land use classification process for two study sites. (**a**) Map of the micro-basin of MACA1 monitoring site; (**b**) Map of the micro-basin of MAAB1. White and black dots are locations of monitoring sites. The yellow line shows the limit of the micro-basins elaborated with GRASS. The blue lines show rivers in the micro-basins, and the green lines show buffer areas drawn around permanent rivers.

A second step included categorizing areas of influence into land use categories (urban settlements, forest, agricultural areas, and grasslands) using satellite imagery. Lacking satellite resolution to define the crop type, this was visually verified on the field (Table 2).

Forest cover in riparian areas was calculated by establishing a buffer of 12 m around permanent streams of each site's influence area. The *i*-Tree Canopy server was used to calculate forest cover within them [27] (Table 2). A total of 12 sites of the 18 were possible to be classified and included in the correlation analysis. Table 2 shows the percentage of each, and the land use in each of the micro-basins that influence the water monitoring sites.

**Table 2.** Coverage percentage of each land use type (crop type, urban, forest, riparian vegetation cover, and grasslands). Note that forest in riparian areas is expressed as forest cover percentage in a buffer area around the micro-basins' permanent rivers. All the other land uses are expressed as a percentage of the micro-basins' total area.

| Site | MACA1 | MACA2 | PAPO1 | PAPO2 | PAPR1 | PAPR2 | MAAB1 | MAAB2 | HARG1 | HARG2 | HASL2 | HASL3 |
|---|---|---|---|---|---|---|---|---|---|---|---|---|
| Area (ha) | 466.9 | 807.8 | 881.4 | 1366.9 | 105.3 | 1355.1 | 832.2 | 1228.0 | 3372.8 | 3657.7 | 37.2 | 61.7 |
| Forest in riparian areas | 39.6% | 38.5% | 9.8% | 6.7% | 21.6% | 7.7% | 63.2% | 79.4% | 56.6% | 67.7% | 23.0% | 53.5% |
| Total agriculture land (includes all crop types) | 52.6% | 38.0% | 67.2% | 65.4% | 57.4% | 56.3% | 10.5% | 15.0% | 8.4% | 9.6% | 67.7% | 62.9% |
| Urban area | 1.4% | 13.2% | 16.0% | 15.7% | 4.6% | 7.9% | 1.3% | 5.8% | 3.5% | 3.9% | 18.5% | 18.6% |
| Forest | 7.7% | 19.5% | 9.6% | 12.0% | 29.9% | 28.5% | 79.4% | 70.2% | 72.4% | 71.9% | 3.0% | 7.3% |
| Grasslands | 33.9% | 20.1% | 7.0% | 6.5% | 8.2% | 6.9% | 8.6% | 8.6% | 15.3% | 14.4% | 7.5% | 9.1% |
| Forest plantations | 4.3% | 3.3% | 0.2% | 0.4% | 0.0% | 0.3% | 0.2% | 0.3% | 0.3% | 0.3% | 3.4% | 2.0% |
| Maize | 32.0% | 24.4% | 31.2% | 32.0% | 14.1% | 21.5% | 7.3% | 7.3% | 4.1% | 4.4% | 19.0% | 14.4% |
| Oats | 9.3% | 10.8% | 18.5% | 13.9% | 6.6% | 11.1% | 2.5% | 4.4% | 2.7% | 2.6% | 5.5% | 8.6% |
| Potato | 6.4% | 3.7% | 11.8% | 12.4% | 33.6% | 18.9% | 0.0% | 0.0% | 0.0% | 0.0% | 0.0% | 0.0% |
| Fava bean | 0.1% | 0.1% | 0.0% | 0.0% | 0.0% | 0.0% | 0.0% | 0.0% | 0.2% | 0.8% | 38.4% | 33.0% |

*2.6. Community Engagement*

Since a major goal of the program was to create awareness and to establish dialogues among final water users and people from the basin, we implemented social community engagement methodologies at each event. Activities implemented included educational sessions, discussion groups, participatory workshops, and guided walks. During data collection on sites, local team leaders also promoted reflections among volunteers about the discussed concepts.

*2.7. Data Analysis*

Two types of data analysis were conducted to understand the characteristics of water quality and the influence of the different land uses and nature-based solutions at each study site. First, the averages and the value of the parameters for each month across the 18 sites were compared to get a general idea of water quality parameters on the different sites and in the basin.

Second, linear regression models were produced between the average of the parameters and the land use areas of the micro-basin for each study site. The models were aimed to identify the specific impact that each land use area has on water quality. Correlations with *p* value < 0.05 were considered statistically significant. All analyses were performed using ggplot2 [28] and psych [29] packages in R [30].

**3. Results**

*3.1. Water Quality in the Basin: Average Values*

Water samples were collected monthly from April 2018 to December 2019. Up to 34 water samples were collected from each site (Table 1) during that period. The analysis allowed us to identify water quality parameters outside of acceptable levels, as well as some seasonality patterns as seen in Table 3. Average values for each parameter are shown in Table 3.

Overall, it was found that average oxygen saturation, alkalinity, *E. coli*, nitrate, nitrite, total phosphorous, total nitrogen, and total suspended solids were outside the acceptable ranges in most of the monitoring sites (Table 3); however, the values were not outside these ranges every month. For example, *E. coli* was higher during warmer months (March–August) and POP and PON were higher during the rainy season (May–September).

**Table 3.** Results of reference and average water quality parameters for each site. (SRP, soluble reactive phosphorus; POP, particulate organic phosphorous; PON, particulate organic nitrogen; DOP, dissolved organic phosphorous; DON, dissolved organic nitrogen; TP, total phosphorus; TN, total nitrogen; TSS, total suspended solids). Average values outside reference levels are highlighted in orange and red.

| | Reference | HARG1 | HARG2 | HASL1 | HASL2 | HASL3 | HOSP1 | HOSP2 | MAAB1 | MAAB2 | MACA1 | MACA2 | PAPO1 | PAPO2 | PAPR1 | PAPR2 | PLTR1 | PLTR2 | SALT |
|---|---|---|---|---|---|---|---|---|---|---|---|---|---|---|---|---|---|---|---|
| Temp. °C | <32 °C [21] | 12.6 | 13.8 | 14.3 | 14.2 | 13.9 | 13.3 | 13.3 | 9.7 | 11.2 | 12.2 | 12.6 | 14.1 | 14.0 | 13.3 | 16.2 | 15.1 | 15.5 | 14.5 |
| pH | 6.5–8.5 [21] | 7.6 | 7.2 | 7.0 | 7.5 | 7.1 | 7.2 | 6.9 | 7.1 | 7.1 | 6.9 | 7.2 | 7.2 | 7.3 | 7.1 | 6.5 | 7.0 | 7.1 | 7.3 |
| O$_2$ ppm | >4 [21] | 6.98 | 6.82 | 6.22 | 6.77 | 6.74 | 6.54 | 6.4 | 7.21 | 6.94 | 6.14 | 6.84 | 6.8 | 6.82 | 7.03 | 5.69 | 6.3 | 6.11 | 6.89 |
| O$_2$ % | 60–125 [21] | 64.9 | 65.15 | 60.11 | 65.24 | 64.63 | 61.91 | 60.99 | 63.07 | 64.71 | 56.83 | 63.64 | 65.36 | 65.31 | 66.27 | 57.09 | 61.82 | 60.51 | 66.94 |
| Turbidity JTU | | 11 | 16 | 6 | 10 | 14 | 19 | 15 | 9 | 15 | 20 | 14 | 16 | 18 | 9 | 16 | 25 | 21 | 32 |
| Alkalinity mg/L | 51–150 [21] | 42 | 46 | 60 | 57 | 58 | 62 | 58 | 48 | 56 | 44 | 46 | 53 | 54 | 32 | 44 | 61 | 60 | 55 |
| Hardness mg/L | 15–200 [21] | 34 | 32 | 34 | 35 | 34 | 37 | 35 | 36 | 37 | 35 | 36 | 40 | 44 | 28 | 36 | 37 | 38 | 39 |
| *E. coli* CFU | <200 and <600 [31] | - | 934 | - | 879 | 427 | 1646 | 7000 | - | 1363 | - | 1185 | - | 1913 | 344 | 426 | 9515 | 17,033 | 7471 |
| Other CFU | | - | 3144 | - | 5383 | 4829 | 1821 | 3272 | - | 1335 | - | 633 | - | 2447 | 1078 | 2079 | 7501 | 9149 | 5212 |
| Flow L/s | | 228 | 238 | 21 | 297 | 52 | 505 | 623 | 6 | 15 | 36 | 39 | 73 | 107 | 63 | 349 | 1482 | 1342 | 674 |
| N-NH$_4^+$ µg/L | <0.5 | 19 | 29 | 155 | 38 | 22 | 19 | 86 | 18 | 15 | 45 | 26 | 35 | 24 | 41 | 50 | 90 | 251 | 23 |
| N-NH$_4^+$ kg/day | | 0.3 | 0.6 | 0.2 | 0.6 | 0.1 | 0.9 | 4.3 | 0.0 | 0.0 | 0.2 | 0.1 | 0.2 | 0.3 | 0.8 | 2.1 | 10.0 | 29.6 | 1.5 |
| N-NO$_3^-$ µg/L | <500 [32] | 654 | 823 | 872 | 831 | 859 | 524 | 577 | 685 | 540 | 906 | 842 | 1275 | 1246 | 922 | 910 | 716 | 716 | 742 |
| N- NO$_3^-$ kg/day | | 10 | 14 | 2 | 23 | 4 | 23 | 31 | 0 | 1 | 3 | 3 | 9 | 11 | 5 | 27 | 90 | 81 | 39 |
| N-NO$_2^-$ µg/L | 90 [33] | 3.8 | 9.0 | 8.0 | 8.5 | 6.5 | 5.5 | 7.1 | 2.3 | 3.1 | 5.5 | 4.0 | 5.7 | 7.8 | 6.1 | 11.4 | 11.1 | 12.8 | 6.7 |
| N-NO$_2^-$ kg/day | | 0.05 | 0.13 | 0.01 | 0.23 | 0.03 | 0.22 | 0.35 | 0.00 | 0.00 | 0.02 | 0.01 | 0.04 | 0.12 | 0.11 | 0.53 | 1.18 | 1.17 | 0.33 |
| SRP µg/L | | 16.8 | 25.3 | 43.6 | 30.8 | 17.1 | 13.5 | 25.5 | 21.1 | 15.9 | 17.9 | 20.1 | 16.2 | 16.2 | 30.9 | 16.4 | 28.6 | 39.5 | 45.9 |
| SRP kg/day | | 0.32 | 0.42 | 0.08 | 0.46 | 0.07 | 0.5 | 1.49 | 0.01 | 0.02 | 0.04 | 0.06 | 0.12 | 0.17 | 0.22 | 0.76 | 3.32 | 3.75 | 1.91 |
| SiO$_2$ µg/L | | 11,309 | 11,609 | 11,844 | 11,957 | 11,826 | 12,666 | 11,999 | 12,453 | 12,133 | 11,956 | 13,195 | 12,116 | 12,142 | 10,458 | 10,609 | 12,504 | 12,421 | 10,663 |
| SiO$_2$ kg/day | | 167 | 191 | 18 | 390 | 50 | 506 | 538 | 7 | 12 | 39 | 41 | 66 | 115 | 73 | 457 | 1274 | 1256 | 452 |
| POP µg/L | | 13.9 | 19.0 | 25.7 | 19.5 | 12.5 | 5.5 | 18.4 | 8.2 | 14 | 14 | 13.8 | 16.9 | 17.4 | 16.7 | 11.6 | 21.7 | 25.9 | 30.8 |
| POP kg/day | | 0.4 | 0.5 | 0.0 | 0.3 | 0.1 | 0.2 | 1.3 | 0.0 | 0.0 | 0.0 | 0.0 | 0.1 | 0.2 | 0.1 | 0.3 | 3.3 | 3.4 | 1.6 |
| PON µg/L | | 132 | 165 | 221 | 242 | 315 | 123 | 150 | 101 | 108 | 115 | 102 | 139 | 136 | 119 | 148 | 169 | 253 | 78 |
| PON kg/day | | 2.9 | 3.8 | 0.3 | 5.1 | 1.4 | 5.5 | 8.1 | 0.1 | 0.1 | 0.4 | 0.3 | 0.8 | 1.2 | 0.7 | 3.1 | 23.4 | 29.3 | 4.4 |
| DOP µg/L | | 23 | 25 | 25 | 23 | 17 | 18 | 23 | 12 | 23 | 22 | 23 | 26 | 25 | 27 | 21 | 31 | 37 | 48 |
| DOP kg/day | | 0.6 | 0.5 | 0.0 | 0.2 | 0.1 | 0.6 | 1.4 | 0.0 | 0.0 | 0.1 | 0.1 | 0.2 | 0.2 | 0.1 | 0.9 | 4.2 | 4.3 | 2.5 |
| DON µg/L | | 200 | 258 | 289 | 302 | 304 | 231 | 233 | 104 | 200 | 232 | 223 | 281 | 241 | 188 | 203 | 266 | 257 | 372 |
| DON kg/day | | 3.5 | 4.7 | 0.4 | 6.1 | 1.3 | 10.5 | 12.8 | 0.1 | 0.2 | 0.7 | 0.7 | 1.8 | 2.0 | 1 | 6.3 | 31.3 | 38.3 | 27.7 |
| TP µg/L | <25 [34] | 54 | 69 | 94 | 73 | 46 | 37 | 67 | 42 | 53 | 53 | 57 | 59 | 58 | 74 | 49 | 81 | 103 | 125 |
| TP kg/day | | 1.3 | 1.4 | 0.2 | 1.0 | 0.2 | 1.4 | 4.2 | 0.0 | 0.1 | 0.1 | 0.2 | 0.4 | 0.6 | 0.4 | 2.0 | 10.8 | 11.4 | 6.0 |
| TN µg/L | <500 and <1000 [34] | 1009 | 1284 | 1545 | 1421 | 1506 | 902 | 1053 | 910 | 867 | 1303 | 1196 | 1736 | 1655 | 1276 | 1323 | 1252 | 1589 | 1223 |
| TN kg/day | | 17 | 23 | 3 | 34 | 7 | 40 | 56 | 1 | 1 | 4 | 4 | 11 | 15 | 7 | 39 | 157 | 179 | 73 |
| TSS g/m$^3$ | <40 [35] | 16.3 | 11.8 | 9.7 | 25.6 | 77.2 | 24.3 | 24.7 | 0.9 | 131 | 20.8 | 19.3 | 23.3 | 21.5 | 9.5 | 5.7 | 22.9 | 21.1 | 33.1 |
| TSS kg/day | | 424 | 372 | 14 | 445 | 458 | 1107 | 1364 | 0 | 16 | 95 | 84 | 124 | 124 | 19 | 72 | 3217 | 2925 | 4316 |

### 3.2. Physical-Chemical Parameters

Water temperature fluctuated over the period of data collection according to the seasons. Temperatures were registered at their lowest from October to January, the coldest month being December 2019 with an average of 10.9 °C, followed by higher temperatures from February until September with the warmest month being May 2018 with 15.8 °C.

The pH average levels were within the acceptable range for all sites (Table 3); however, sites HASL3, HOSP2, MACA1, MACA2, and PAPR2 presented one or two months with pH values lower than the reference. The average pH was relatively stable throughout the monitoring period, with slightly more acidic values during the rainy season, from May 2019 to September 2019.

The average dissolved oxygen values were within the reference levels for all sites, except for HOSP2 and PAPR2 sites, where less than 4 ppm was observed in one month. Sites MACA1 and PAPR2 had lower average values of oxygen saturation than the references. Oxygen saturation was below the reference levels in sites MACA1 and PAPR2.

Sites HARG1, HARG2, MAAB1, MACA1, MACA2, and PAPR2 presented lower alkalinity values than the reference levels.

### 3.3. Bacteriological Parameters

Average *E. coli* concentration levels were much higher than the recommended limit in all sites. Levels were even higher during warmer months (March–August). While there is no reference level for other coliforms, their average concentration was extremely high as compared to *E. coli* reference levels, ranging from 633 to 9149 CFU/100 mL. The site in the river after the discharge of the wastewater treatment plant had an average value for *E. coli* of 17,033 CFU/100 mL. This was almost double the average value of the river before the discharge, and around 16 times more than the monitoring sites above the main human settlements.

### 3.4. Water Nutrient Content

Nitrogen from nitrates ($N-NO_3^-$), total phosphorus, and total nitrogen concentration average values were above the reference levels for all the sites. Phosphate concentration average levels were higher from the months of February to May. Particulate organic phosphorus (POP) and particulate organic nitrogen (PON) values were higher during rainy season. For total phosphorus concentration, all sites showed eutrophic (24–96 μg/L) or hypereutrophic (>96 μg/L) levels, and for total nitrogen concentration, all sites showed mesotrophic (500–1000 μg/L) or eutrophic (1000–2000 μg/L) levels. Average levels (averaging all studied months) of total suspended solids ranged between excellent (≤25 mg/L) and good (>25 y ≤75 mg/L) [34] in all sites; however, there were months in the middle of the rainy season (July and August) when most sites overpassed eutrophication levels, reaching, in some cases, up to 288 mg/(MACA2).

### 3.5. Correlation between Land Use and Water Quality Parameters

The results of linear regression models between the different land use values and water quality parameters showed significant correlations. Each land use correlated to a different set of water quality parameters. All significant correlations can be seen in Table 4.

Urban settlements correlated with higher levels of alkalinity, PON and DON, total nitrogen, total solids, hardness, and *E. coli*. Agriculture correlated with higher temperature, total nitrogen, $N-NO_3^-$, $N-NH_4^+$, turbidity, and hardness.

When analyzed separately, all agricultural land uses correlated with a different set of water quality parameters. Maize and oats correlated with higher levels of turbidity, hardness, $N-NO_3^-$, and total nitrogen. Fava bean and potato cultivation correlated with higher values, showing worsened conditions in water quality. Fava bean correlated with non-*E. coli* coliforms, POP, PON, DON, and total suspended solids. Potato cultivation correlated with higher levels of temperature, $N-NO_3^-$, $N-NO_2^-$, $N-NH_4^+$, DOP, silicates, and with lower alkalinity and dissolved $O_2$ levels.

**Table 4.** Significant correlations between land use and water quality parameters' values. Three significance levels were considered in this study. Correlations marked with "*" have a *p* value lower than 0.05; those marked with "**" have a *p* value lower than 0.01, and those marked with "***" have a *p* value lower than 0.001.

| Practice | Land Use Variable | Water Parameter | Slope | Adjusted R$^2$ | *p* |
|---|---|---|---|---|---|
| Forest cover | Percentage of catchment area with forest cover | TN (µg/L) | −702.30 | 0.602 | 0.002 ** |
| | | N-NO$_3^-$ (µg/L) | −473.0 | 0.409 | 0.015 * |
| | | DON (µg/L) | −119.16 | 0.375 | 0.020 * |
| | Forest area (Ha) | POP (kg/day) | $1.215 \times 10^{-4}$ | 0.499 | 0.006 ** |
| Forest cover in riparian buffer areas | Percentage of buffer area with forest cover | Temp. (°C) | −4.452 | 0.403 | 0.016 * |
| | | TN (µg/L) | −800.3 | 0.505 | 0.006 ** |
| | | N-NO$_3^-$ (µg/L) | −696.33 | 0.632 | 0.001 ** |
| | Buffer area with cover (Ha) | POP (kg/day) | 0.008 | 0.377 | 0.02 * |
| Agriculture | Percentage of catchment area with agricultural use | Temp. (°C) | 4.206 | 0.32 | 0.032 * |
| | | TN (µg/L) | 920.62 | 0.649 | <0.001 *** |
| | | N-NO$_3^-$ (µg/L) | 659.08 | 0.514 | 0.005 ** |
| | | N-NH$_4^+$ (µg/L) | 27.884 | 0.286 | 0.042 * |
| | Agricultural area (Ha) | Turbidity (JTU) | 0.008 | 0.33 | 0.030 * |
| | | Hardness (mg/L) | 0.009 | 0.404 | 0.016 * |
| | | N-NO$_3^-$ (µg/L) | 0.477 | 0.353 | 0.025 * |
| Grasslands | Percentage of catchment area with grasslands | - | | | |
| | Grassland area (Ha) | POP (kg/day) | 0.001 | 0.497 | 0.006 ** |
| Urban settlements | Percentage of cachment area with urban settlements | Alkalinity (kg/L) | 77.165 | 0.399 | 0.016 * |
| | | PON (mg/L) | 602.08 | 0.316 | 0.033 * |
| | | DON (µg/L) | 632.50 | 0.503 | 0.006 ** |
| | | TN (µg/L) | 2948.5 | 0.458 | 0.009 ** |
| | | Total solids (g/m$^3$) | 183.605 | 0.324 | 0.031 * |
| | Urban settlements area (Ha) | Hardness (mg/L) | 0.035 | 0.321 | 0.032 * |
| | | *E. coli* (CFU/100 mL) | 5.239 | 0.435 | 0.045 * |
| Maize | Percentage of cachment area with maize cultivation | Turbidity (JTU) | 20.372 | 0.275 | 0.046 * |
| | | TN (µg/L) | 1790.9 | 0.444 | 0.011 * |
| | | N-NO$_3^-$ (µg/L) | 1599.36 | 0.594 | 0.002 ** |
| | Maize cultivation area (Ha) | Turbidity (JTU) | 0.019 | 0.382 | 0.019 * |
| | | Hardness (mg/L) | 0.022 | 0.504 | 0.006 ** |
| | | N-NO$_3^-$ (µg/L) | 1.057 | 0.364 | 0.022 * |
| Oats | Percentage of catchment area with oat cultivation | Hardness (mg/L) | 47.701 | 0.294 | 0.040 * |
| | | TN (µg/L) | 4207.7 | 0.557 | 0.003 ** |
| | | N-NO$_3^-$ (µg/L) | 3721.62 | 0.723 | <0.001 *** |
| | Oat cultivation area (ha) | Turbidity (JTU) | 0.034 | 0.283 | 0.043 * |
| | | Hardness (mg/L) | 0.042 | 0.414 | 0.014 * |
| | | N-NO$_3^-$ (µg/L) | 1.992 | 0.305 | 0.036 * |
| Fava bean | Percentage of cachment area with fava bean cultivation | Other coliforms (CFU/100 mL) | 9713.1 | 0.768 | 0.003 ** |
| | | Total solids (g/m$^3$) | 99.208 | 0.422 | 0.013 * |
| | | Total solids (kg/day) | 799.13 | 0.340 | 0.027 * |
| | | PON (µg/L) | 428.832 | 0.791 | <0.001 *** |
| | | DON (µg/L) | 260.69 | 0.333 | 0.029 * |
| | Fava bean cultivation area (Ha) | Total solids (kg/day) | 13.104 | 0.598 | 0.002 ** |
| | | POP (kg/day) | 0.009 | 0.341 | 0.027 * |
| | | PON (µg/L) | 4.095 | 0.379 | 0.020 * |
| | | PON (kg/day) | 0.102 | 0.349 | 0.025 * |

**Table 4.** *Cont.*

| Practice | Land Use Variable | Water Parameter | Slope | Adjusted $R^2$ | *p* |
|---|---|---|---|---|---|
| Potato | Percentage of catchment area with potato cultivation | Alkalinity (kg/L) | −43.633 | 0.308 | 0.036 * |
| | | N-NH$_4^+$ (μg/L) | 67.105 | 0.315 | 0.034 * |
| | | SiO$_2$ (μg/L) | −4538.4 | 0.338 | 0.028 * |
| | Potato cultivation area (Ha) | Dissolved O$_2$ (ppm) | −0.003 | 0.376 | 0.020 * |
| | | Temp. (°C) | 0.013 | 0.364 | 0.022 * |
| | | N-NO$_3^-$ (μg/L) | 1.521 | 0.282 | 0.044 * |
| | | N-NO$_2^-$ (μg/L) | 0.019 | 0.317 | 0.033 * |
| | | N-NO$_2^-$ (kg/day) | 0.001 | 0.475 | 0.008 ** |
| | | N-NH$_4^+$ (kg/day) | 0.005 | 0.416 | 0.014 * |
| | | DOP (kg/day) | 0.002 | 0.266 | 0.050 * |

*3.6. Correlation between Nature-Based Solutions and Water Quality Parameters*

Total forested area and riparian cover correlated to better levels of water quality parameters (Table 4). Total forested area correlated with lower levels of N-NO$_3^-$, DON, and total nitrogen, and with higher levels of POP. Riparian cover correlated with lower temperature, N-NO$_3^-$, and total nitrogen, and with higher levels of POP. On the other hand, grasslands only correlated with higher levels of POP.

**4. Discussion**

The results show signs of pollution and eutrophication on sites in the middle and upper basin. One of the most alarming results of the research was that the average *E. coli* CFU was much higher than the recommended 200 CFU/100 mL to be safe for human contact and to protect water life [31]. The sites with higher CFU were sites located downstream to human settlements. This reflects that a portion of wastewater goes into the rivers untreated. In addition, the site in the Amanalco river after the discharge of the wastewater treatment (PLTR2) plant had higher levels (17,033 CFU/100 mL) of *E. coli* than the site before the plant's water discharge (9515 CFU), which shows that the plant's discharge is polluting the Amanalco river; hence, the treatment plant is not working adequately or at all. Depending on the strain and transmission method, *E. coli* can have severe effects on human health. If accidentally ingested, it can cause foodborne diseases that can be lethal, especially for children and the elderly. Infection can also lead to the development of haemolytic uraemic syndrome (HUS)—which causes renal failure—haemolytic anaemia, thrombocytopenia, and neurological complications such as seizure, stroke, and coma [36].

The concentration of nutrients and suspended solids in the water indicate eutrophic and hypereutrophic conditions according to phosphorus concentrations, and mesotrophic and eutrophic conditions according to nitrogen concentrations for temperate stream types [37]. This is a combined effect of wastewater, non-point pollution sources such as fertilizers and pesticides used in agriculture, and, possibly, the effect of other activities not included in this study, such as trout production [6,10,11]).

The impact of the sources of pollution can also be seen in two other results from this study. One-third of the sites (HARG1, HARG2, MAAB1, MACA1, MACA2, and PAPR2) showed levels of alkalinity that were lower than the references for ecological health. Alkalinity is depleted from water bodies when acid pollutants are added. This can increase vulnerability of water ecosystems because it reduces the water bodies' buffering capacity to acid pollutants, and increases the effect of these pollutants on the water bodies' pH. The other effect associated with high levels of nutrients seen in the results is low levels of dissolved oxygen, which is consumed by microorganisms during organic matter decomposition and remineralization of both in situ and allochthonous organic matter [37,38].

Nutrient loads were estimated considering the total water flow in each site. This showed that the total nutrients load and total suspended solids load were lower on sites in the upper basin and higher in the lower basin. These findings may be due to higher water volumes in the monitoring sites located downstream, with a greater influence from various

land uses and populations, as has been observed in larger watersheds [9,39]. The "El Salto" site, located in the lowest part of the monitoring area and near the most populated village of Amanalco, reached up to 4316 kg of total suspended solids per day. This reflects high levels of erosion and runoff in the basin due to unsustainable agricultural and other land use practices.

Correlations between land use and water quality parameters allowed for an understanding of the different impacts of each crop system and of the forest area and human settlements over water. For example, human settlements showed a correlation with higher levels of alkalinity, which can be related to the discharge of detergents in gullies and streams [40]. Human settlement also had a positive correlation with higher levels of hardness (total content of calcium and magnesium), total nitrogen, and total solids, and was the only land use that correlated to higher levels of *E. coli*, reflecting the impact of wastewater discharges in the river.

Agricultural practices showed, in general, a negative impact on water quality, having a positive correlation with higher levels of turbidity, hardness, total nitrogen, nitrates, and ammonium, confirming the findings of Gorgoglione et al. [5], Nazari-Sharabian et al. [2], Kandler et al. [3], Nepomuscene Namugize et al. [4], and Mirsaeedghazi [6]. This reflects the use of fertilizers which end up in water bodies through diffuse pollution, as well as soil erosion and runoff caused by tilling and poor irrigation practices as observed elsewhere [41,42]. However, the correlations of specific crops in the Amanalco-Valle de Bravo watershed showed that the impact on water quality varies depending on the type of agricultural product due to their production systems. Maize and oats are the crops with a lower impact on water quality, only showing a correlation with higher levels of turbidity, hardness, total nitrogen, and N-NO$_3$$^-$.

On the other hand, fava bean cultivation correlated with higher levels of DON and total suspended solids.

Potato was the crop that showed the most negative impact on water quality. This may be due to the use of agrochemicals and fertilizers (Supplementary Materials Tables S1 and S2). The high use of fertilizers for potato cultivation is reflected in the correlation of this crop with higher levels of, N-NH$_4$$^+$, N-NO$_3$$^-$, and N-NO$_2$$^-$ [32,33,43]. The impacts on water by potato cultivation also correlates with lower levels of alkalinity. When an acid is added to water, hydrogen ions combine with carbonate and bicarbonate ions. This reaction prevents acids from changing the water pH, but reduces the alkalinity concentration [21,44]. Additionally, the potato cultivation area correlated with lower levels of dissolved oxygen in the water, which is consumed when there are higher levels of organic matter in the water, and higher TSS (particularly in 2018) due to straight-line planting in rows parallel to the slope to increase runoff and reduce humidity, which causes extremely high levels of soil erosion.

Dissolved forms of nutrients in water bodies are very bioavailable and have a fast effect on eutrophication processes [38,45]. Fava bean correlated with higher levels of other coliforms, which might be explained by the intensive use of manure as fertilizer for this crop. It also correlated with POP and PON. Particulate forms of nutrients originate from the tissues of living organisms from aquatic ecosystems, or from organisms from terrestrial ecosystems that were dragged by superficial water, which have started slowly decomposing. Particulate forms will be degraded into dissolved forms of nutrients, and then will become bioavailable.

Grasslands, which are native ecosystems in some areas of the basin, and in other areas are the result of land use change, showed a significant correlation with POP, which is related to slow decomposition of organic matter.

While correlations between land use and water quality parameters allowed for confirmation of the impact that different land uses have on water quality in the basin, they also allowed for the identification of land uses that correlate with better water quality parameters, and that can be used as nature-based solutions to mitigate the impact to water bodies. The first one is the total forest area in the micro-basin. Forest area correlated with

lower levels of total nitrogen and lower levels of nitrates. Sites with a higher proportion of forest area have a lower proportion of area destined for alternative uses in the basin, such as agriculture, grazing, and urban settlements, and thus are less exposed to pollution derived from these uses, which partly explains the higher water quality of these sites. Additionally, better water quality in these sites reflects the capacity of forests to provide ecosystem services, such as water quality regulation and protection from soil erosion [46], and the role of terrestrial vegetation in the uptake of bioavailable phosphorus [5]. Similar results have been found in other studies, such as Gorgoglione et al., 2020 [5] in an Uruguayan basin.

Forest cover in riparian areas, besides correlating with lower levels of total nitrogen and nitrates, also correlates with lower temperatures, lower levels of ammonium, and lower levels of total phosphorous, showing the potential of restoring forest cover in riparian areas to protect streams and water quality from land use impacts and pollutants. Besides, forests in riparian areas protect water courses because they are located in between them and agriculture. This allows them to work as buffers, retaining pollutants and stopping runoff to get to water bodies [47]. Furthermore, shade over water bodies protects them from extreme weather and maintains a lower water temperature, improving ecological conditions and slowing down eutrophication processes. These results confirm what Babaei et al., 2019 found in a basin in Iran, where the use of filter strips with vegetation between water bodies and cropland reduced the nitrate concentration and TN [6].

Due to this, restoration of forests in riparian areas is an effective way of protecting water bodies from detrimental pollutants and increasing water quality, thus enhancing aquatic ecosystems, reducing risks to human health, and reducing costs associated to water purification for human use.

## 5. Conclusions

The research project allowed us to determine the deterioration of water quality starting from the upper basin and down to the middle part of the basin. *E. coli*, total nitrogen, total phosphorous, and $N-NO_3^-$ levels were above the levels recommended for the protection of aquatic life and human contact in most of the sites. This demonstrates the need to implement corrective measures starting from the upper basin.

This study also allowed us to identify how different land uses impact specific water quality parameters, shedding light on the mechanisms of water pollution and the adequate measures that are required to mitigate them. According to the results, the following measures are recommended to reduce the impacts and restore the ecosystem services:

1. To prevent the impact from wastewater discharge, it is necessary to install efficient water treatment technologies, because, currently, most of the basin's creeks are polluted with *E. coli* and other coliforms.
2. The creation or strengthening of public policies and economic instruments such as payments for ecosystem services seeks to promote agricultural best management practices, reduce the use of agrochemicals, and conserve forests.
3. Specific measures should be taken to regulate the expansion of potato cultivation and to control the management practices that are used in that crop. This can be attained by working with local farmers to prevent them from renting their land, and to engage them in organic and environmentally friendly potato production that can reach organic markets in Valle de Bravo in the short term, and at a regional level in the medium term, by organizing themselves as cooperatives of organic/environmentally friendly potato producers.
4. Promoting the conservation and restoration of forest cover in riparian areas to protect streams and water quality from the land use impact. This will need a specific campaign and investment to engage farmers that own land near creeks and rivers.
5. Mechanisms to protect and restore total forest cover in the basin, such as sustainable community forest management for timber production, forest vigilance, or ecotourism, which should be generated or supported to increase water quality regulating services and improve water quality in the basin.

Finally, although citizen-based science methodology as well as participatory workshops between local actors and visitors were costly and required high levels of logistics, both allowed for capacity building, raising awareness on the importance of water quality and nature-based solutions, and dialogue between upper watershed inhabitants and city water users. This helps to advocate for better basin management policies and to increase the feasibility of market mechanisms, such as PES schemes.

**Supplementary Materials:** The following are available online at https://www.mdpi.com/article/10.3390/su131910519/s1. Table S1: Agricultural cycles in the Amanalco-Valle de Bravo Basin. Table S2: Agricultural inputs reported for each crop in the Amanalco-Valle de Bravo basin.

**Author Contributions:** Conceptualization and methodology, J.C.C., L.M.R., J.R.Z., S.M.S.d.T., M.M.I. and A.C.T.; formal analysis, J.C.C., L.M.R. and J.R.Z.; investigation, J.C.C.; data curation, J.C.C., L.M.R., J.R.Z., J.D.V. and M.M.I.; original draft preparation, J.C.C.; writing—review and editing, L.M.R., J.R.Z. and S.M.S.d.T. All authors have read and agreed to the published version of the manuscript.

**Funding:** This research was funded by Earthwatch Institute through the HSBC Water Programme and developed by the Mexican NGO, Consejo Civil Mexicano para la Silvicultura Sostenible.

**Institutional Review Board Statement:** Not applicable.

**Informed Consent Statement:** Not applicable.

**Acknowledgments:** We would like to thank Olivia Faustino Zárate, Olivia Francisco Cipriano, Juan Sotero Aviles, and Zeferino Espinoza Eugenio for their participation in teaching the citizen scientists; Macarena Cárdenas for her valuable contributions with reviewing the manuscript; Sarai Zelaida and Erick Ricardo Hjort Colunga for their support and contributions in the development of the project; and HSBC volunteers participating in this project for their contributions monitoring water quality, and their readiness to participate in collective discussions and the design of personal and corporative sustainability strategies. We appreciate the assistance of F. Sergio Castillo Sandoval, who carried out all of the analyses at the Aquatic Biogeochemistry Laboratory, UNAM.

**Conflicts of Interest:** The authors declare no conflict of interest. The funders had no role in the design of the study; in the collection, analyses, or interpretation of data; in the writing of the manuscript; or in the decision to publish the results.

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
