# Peer review of "Evaluation of the Impacts of Land Use in Water Quality and the Role of Nature-Based Solutions: A Citizen Science-Based Study"

_sustainability, doi:10.3390/su131910519_

Round 1

Reviewer 1 Report

The manuscript "Evaluation of the impacts of land use in water quality and the role of nature-based solutions: a citizen-science based study" is scientifically sound, and is well-written.

However, the authors should consider the following comments:

- Line 76 until the end of the paragraph: acknowledgments do not have to be mentioned here. Please move them to the end of the manuscript and put them in the Acknowledgment section.

- Line 227: table 3 --> Table 3

- At several locations throughout the manuscript, E. coli is written as italic, while in some locations it is nonitalic. Please be consistent.

- Line 268: august --> August

- The authors are encouraged to use the following articles in their manuscript, where appropriate, to increase the quality of the discussion:

  • Water Quality Modeling of Mahabad Dam Watershed–Reservoir System under Climate Change Conditions, using SWAT and System Dynamics. Water, 2019, 11, 394. DOI: 10.3390/w11020394
  • Surface Runoff and Pollutant Load Response to Urbanization, Climate Variability, and Low Impact Developments – A Case Study. Water Supply, 2019, 19(8), 2410-2421. DOI: 10.2166/ws.2019.123
  • Identification of Critical Source Areas (CSAs) and Evaluation of Best Management Practices (BMPs) in Controlling Eutrophication in the Dez River Basin. Environments, 2019, 6, 20. DOI: 10.3390/environments6020020
  • Climate Change and Eutrophication: A Short Review. Engineering, Technology & Applied Science Research, 2018, 8(6), 3668-3672. DOI: 10.48084/etasr.2392

Author Response

Thank you very much for your time and your valuable contributions, they were very helpful to improve the manuscript.

Please find the point-by-point response in the attached file.

Reviewer 2 Report

Please, see the enclosed file.

Author Response

(The authors gave the same response as above.)

Reviewer 3 Report

Evaluation of the impacts of land use in water quality and the role of nature-based solutions: a citizen-science based study

-The subject of this paper is suitable for publication profile of Sustainability.
-The obtained research results may be the basis for publication, but they should be supplemented in the future and should constitute an introduction to further research in this field.
- The novelty of the work is not clearly explained. I would like to suggest to the authors to define better the aim and to point out the novelty of the work. The most important achievement result from the conducted research with regard to the previously published results should be emphasized.
-The chemical formulas throughout the manuscript lack subscripts and superscripts.
- In the discussion of the research results, reference should be made to the relevant literature. Please provide relevant comparisons.
- Conclusions is recommended to contain more numerical data emphasizing the more important results.
- In future studies, the authors of the manuscript should use total organic carbon (TOC) and dissolved organic carbon (DOC) as an indicator of organic matter.

Author Response

(The authors gave the same response as above.)

Round 2

Reviewer 2 Report

I am glad to see that the authors followed all my suggestions and recommendations. In my opinion, the manuscript was substantially improved and it is ready to be published.

Reviewer 3 Report

The article has been greatly improved by the authors.

I am satisfied with the modifications added to this paper, and also with the improvements.

The manuscript may be published in the journal Sustainability  in present form.